

# Joint Analysis of Convective Structure from the APR-2 Precipitation Radar and the DAWN Doppler Wind Lidar During the 2017 Convective Processes Experiment (CPEX)

5   F. Joseph Turk[1], Svetla Hristova-Veleva[1], Stephen L. Durden[1], Simone Tanelli[1], Ousmane Sy[1], Dave Emmitt[2], Steve Greco[2], Sara Q. Zhang[3]

[1]Jet Propulsion Laboratory, California Institute of Technology, Pasadena CA 91107 USA
[2]Simpson Weather Associates, Charlottesville VA 22902 USA
[3]Global Modeling and Assimilation Office (GMAO), Goddard Space Flight Center, Greenbelt MD 20771 USA

10  *Correspondence to*:  F. Joseph (Joe) Turk (jturk@jpl.caltech.edu)

**Abstract.** The mechanisms linking convection and cloud dynamical processes is a major factor in much of the uncertainty in both weather and climate prediction.  Further constraining the uncertainty in convective cloud processes linking 3-D air motion and cloud structure through models and observations is vital for improvements in weather forecasting, and understanding limits on atmospheric predictability.   To date, there have been relatively few airborne observations 15  specifically targeted for sampling convective cloud processes linking 3-D air motion and transport of water vapor near clouds, and the subsequent development (or non-development) of convective precipitation.  During the May-June 2017 Convective Processes Experiment (CPEX), NASA DC-8-based airborne observations were collected from the JPL Ku/Ka-band Airborne Precipitation Radar (APR-2) and the 2-um Doppler Aerosol Wind (DAWN) lidar during approximately 100 flight hours.  Frequent dropsonde data accompanied the DAWN observations for validation purposes, and to provide 20  complement wind profiles in and near convection. For CPEX, the APR-2 provided vertical air motion and structure of the cloud systems in nearby precipitating regions where DAWN is unable to sense.  Conversely, DAWN sampled vertical wind profiles in aerosol-rich regions surrounding the convection, but is unable to sense the wind field structure within most clouds.  In this manuscript, the complementary nature of these data are presented from the June 10-11 flight dates, including the APR-2 precipitation structure and Doppler wind fields, and adjacent wind profiles from the DAWN and dropsonde data.

## 1       Introduction.

The mechanisms linking convection and cloud dynamical processes is a major factor in much of the uncertainty in both weather and climate prediction.  The associated mesoscale convective systems (MCS) produce much of the Earth's rainfall and are responsible for the bulk of the heat and moisture transport from the Earth's surface into the upper troposphere.  The cold pool dynamics are thought to be an important mechanism to facilitate the development of MCSs in the tropical 30  atmosphere (*Chen et. al.*, 2015; *Zuidema et. al.*, 2017), as well as interactions between individual isolated convective storms



(*Raymond et. al.,* 2015). These atmospheric boundaries can have significant impact on deep convection, affecting its initiation, updraft strength and longevity. The intensity and size of the cold pools is strongly dependent upon the vertical distribution of the temperature and humidity and the vertical shear of the horizontal wind. While the overall processes responsible for these interactions have been identified for some time, their precise nature and interactions remains under-

constrained by observations, due to the difficulty in obtaining accurate, vertically resolved pressure, temperature, wind and water vapor in the proximity of developing convective clouds. Moreover, increasing evidence points to control of convection by the relatively smaller and more variable amount of moisture above the boundary layer, in the free troposphere (*Schiro and Neelin*, 2019). Further constraining the uncertainty in convective cloud processes *linking 3-D air motion and cloud structure* through models and observations is vital for improvements in weather forecasting and understanding limits

on atmospheric predictability.

The resolution of the precipitation radar onboard the Tropical Rainfall Measuring Mission (TRMM; 1997-2014) and the subsequent Global Precipitation Measurement (GPM; 2014-current) missions (4-km horizontal resolution; 250-m vertical) have enabled numerous observational-based studies of MCS convective structure and features (*Jiang et. al.*, 2011).

However, the associated dynamical (air motion) wind field associated with MCS features at this scale not well-represented by current space-based wind profile observing capabilities. The majority of available atmospheric wind observations are primarily water vapor and cloud-tracked atmospheric motion wind vectors (AMV) derived from operational geostationary satellites (*Velden et. al*, 2005), which can be refreshed as quickly as 15-minutes, but are mainly indicative of large-scale mid-to-upper level air motion patterns. Observations of wind vectors in the periphery of smaller-scale cloud systems, especially

in the 2-km nearest the Earth (the approximate delineation of the boundary layer) are much less abundant. Outside of ground-based profiling networks, very few over-ocean wind profile observations at TRMM/GPM-like horizontal resolution are available.

A space-based Doppler wind lidar (DWL) capability has been envisioned as one means to overcome this observational

shortcoming (*Baker et. al.*, 2014). Over the past decade, airborne DWL field campaigns have been conducted (*Lux et. al.*, 2018), recently in preparation for the deployment (August 2018) of the first-ever spaceborne DWL, the Atmospheric Dynamics Mission (ADM-Aeolus) of the European Space Agency (ESA) (*Stoffelen et. al.*, 2005). Aeolus provides vertical profiles of the line-of-sight (LOS) winds from its (spacecraft) nadir angle of near 35-degrees, with at an approximate 100-km horizontal resolution and 200-km separation between profiles. Since a main application of Aeolus is to numerical weather

prediction data assimilation (*Horányi et. al.*, 2015), observations from recent campaigns with a DWL such as the THORPEX Pacific Asian Regional Campaign (TPARC) were largely focused towards improvement of tropical cyclone forecasts (*Pu et. al.*, 2010). These airborne campaigns have validated the capabilities of a DWL to provide wind profiles in the boundary layer (*Bucci et. al.*, 2018; *Zhang et. al.*, 2018). There has been relatively less focus in collection and analysis of airborne DWL observations in relation to the convective processes linking air motion and transport of water vapor near clouds, and



the subsequent development (or non-development) of convection.   One main reason is that previous campaigns often lacked nadir scanning Doppler precipitation radar capabilities on the same aircraft to enable matched radar-DWL observations.  A scanning precipitation radar provides the actual 3-D representation of the condensed water mass field, and the vertical Doppler winds and associated microphysical vertical structure (*Rowe and Houze*, 2014; *Rowe et. al.*, 2012).  These data provide one means to validate the forecasted model precipitation structure (e.g., presence/absence of convection, timing,

location), that results when the DWL wind vectors are assimilated into cloud resolving models.

In this manuscript, airborne DWL and Doppler precipitation radar observations are presented from the NASA-sponsored Convective Processes Experiment (CPEX), which took place between 25 May and 24 June 2017, based out of Fort Lauderdale, FL.  The goals of CPEX were to improve the understanding of convective processes during initiation, growth,

and dissipation, using a combination of observations and cloud-resolving models.  In particular, to measure what combinations of environmental structure and observed convective properties such as vertical velocity and reflectivity profiles, result in rapid upscale growth of a convective system into a large organized mesoscale convective system (MCS), or alternatively, result in failure to grow or rapid decay.  This manuscript will describe and present only the airborne precipitation radar and DWL observations; a separate manuscript will present the associated mesoscale model simulations

and DWL data assimilation experiment results (*Zhang et. al.*, 2019).

## 2       CPEX Overview.

During CPEX, sixteen NASA DC-8 airborne missions were flown into the Gulf of Mexico, Caribbean Sea and the Atlantic Ocean. Each date is summarized in Table 1. During each flight, joint observations were collected from the JPL Ku/Ka-band Airborne Precipitation Radar (APR-2)[1] and the 2-um Doppler Aerosol Wind (DAWN) DWL, covering a variety of isolated,

scattered, and organized deep convection, totalling approximately 100 flight hours.  Frequent dropsonde data accompanied the DAWN observations for validation purposes, and to provide complementary wind profiles near convection.  The dropsondes system used during CPEX was the High Definition Sounding System (HDSS) dropsonde delivery system developed by Yankee Environmental Services (*Black et. al.*, 2017).   While the dropsonde data are shown on several of the following figures, they are not discussed in this manuscript.


| Flight | Date | Observations |
|:---:|:---:|:---|
| 1 | 27 May 2017 | First local science flight; box pattern in central Gulf; clear air only. |
| 2 | 29 May 2017 | Sampling of scattered convection in NW Caribbean; cells at 1813, 1942-2000. |
| 3 | 31 May 2017 | Multiple boxes over Atlantic, near Bahamas and north of Hispaniola; mostly clear but cells |

---

[1] In 2015, APR-2 was augmented with an additional W-band (94 GHz) Doppler radar for an expanded APR-3 capability.  Owing to logistical details, the W-band radar was unavailable for CPEX in 2017, hence the use of the APR-2 system.



| | | |
|---|---|---|
| | | at 1936, 2120. |
| 4 | 1 June 2017 | Convective system over eastern Gulf; multiple passes over convection. 25-min data loss at Ka-band due to TWT amplifier breaker trip. |
| 5 | 2 June 2017 | Extended E-W box over western and central Gulf; clear areas and some convective cells, for example at 1750, 1928; decaying convection between 1830-1900, 2100-2110. |
| 6 | 6 June 2017 | Convection over eastern Gulf, especially near 1858, 1955-2115, 2105, 2140. |
| 7 | 10 June 2017 | Boxes east of the Bahamas; stratiform with some convection on ascent between 1840-1850, small cells in box 1925, 2004, 2035-2045, 2118, 2140, 2210-2216. |
| 8 | 11 June 2017 | E-W legs over convective system in central Gulf; isolated cells at 1801, 1830, 1850; extensive precipitation on lines starting at 1900, 1920, and N-S line starting 2005. |
| 9 | 15 June 2017 | Caribbean, east of Yucatan; convection near 1920, 1940, 1953, 2010. |
| 10 | 16 June 2017 | Caribbean, boxes east of Yucatan; convection near 1830-1940, 2050-2140. |
| 11 | 17 June 2017 | Caribbean, boxes east of Yucatan; convective cells at 1745, 1800-1815, 2044-2054, 2223; sampled convective system with box pattern between 1900 and 2030. |
| 12 | 19 June 2017 | E-W legs over north-central and northeast Gulf of Mexico, Tropical Storm Cindy; extensive precipitation between 1700-1820, 1840-2005; numerous isolated cells to 2130, then more extensive areas to 2224. |
| 13 | 20 June 2017 | Bow-tie pattern in central Gulf of Mexico; convective system between 1742-1754, cells 1815-1820, very shallow convection 1923, extensive precipitation between 2110-2150. |
| 14 | 21 June 2017 | E-W flight across Gulf of Mexico; isolated cells at 1842, 1942, 2028, 2107, 2124, 2158, 2240; stratiform/transitional between 1925-1937. |
| 15 | 23 June 2017 | Box pattern to east of Bahamas; crossed isolated cells at 1832, 1859, 1910, 1917; multiple lines over area with isolated cells between 1912-1939. |
| 16 | 24 June 2017 | Over and around Cuba; convection at 1744, box pattern cells near 1829, isolated cells 1843-1944; mature cell near 2106, more cells 2112-2143. |

**Table 1. Summary of CPEX flight dates.**

APR-2 is a 2-frequency Doppler radar, originally developed as an airborne prototype for the second-generation GPM/DPR precipitation radar (*Sadowy et al*., 2003). The APR-2 has flown in numerous airborne field campaigns outside of CPEX, most recently the ORACLES (2016-2018) and CAMP2Ex (2019) campaigns. APR-2 acquires simultaneous measurements of multiple parameters at both Ku- and Ka-band (14 and 35 GHz, respectively), including co- and cross-polarized radar backscatter, and LOS Doppler velocities of hydrometeors, with a maximum unambiguous velocity of ±27.5 (Ku-band) and ±10.4 (Ka-band) m s$^{-1}$. From a nominal 10-km flight altitude, the horizontal resolution at the surface is ~800-m, with a vertical range resolution and sampling of 50- and 30-m (slightly oversampled). Based upon analysis of radar surface backscatter measurements from CPEX, the reflectivity calibration is accurate to within 1-2 dB. From these basic measurements, APR-2 can depict the cloud macroscopic structure (extent, vertical air motion) and estimate the microphysical structure (water content, precipitation intensity, hydrometeor size distribution) of the associated precipitation



(*Durden et. al.*, 2012). These resolutions are adequate to capture cloud features down to the resolution typical of high-
resolution cloud models, and appropriate for comparison with DAWN wind profiles in the vicinity near isolated, scattered,
and organized deep convection.

DAWN is NASA's highly capable airborne DWL with a 2-micron laser that pulses at 10 Hz (*Kavaya et. al.*, 2014). It has
previously participated in the NASA GRIP (2010) and Polar Winds (2014-15) airborne campaigns. DAWN can provide
high resolution (4-12 km in the horizontal and 35-150 m in the vertical) wind measurements in clear as well as partly cloudy
conditions. The lidar gathers data in a conical pattern at a constant 30° elevation angle, and collects LOS wind profiles at up
to five azimuth angles located at -45°, -22.5°, 0°, 22.5° and 45° relative to the aircraft flight direction (Figure 1). During
CPEX, DAWN also collected LOS data at only two azimuth angles, -45° and 45°. Since these LOS wind profiles view the
local wind field from multiple azimuth angles, these LOS profiles are analyzed to estimate the vertical profile of the
horizontal wind components (*u, v*) at different pressure levels using the Adaptive Signal Integration Algorithm (ASIA)
processing (*Kavaya et. al.,* 2014). DAWN data are available in both the native LOS format, and processed wind vector (*u, v*)
profile format. In this manuscript, the profile data are used to evaluate the wind field near clouds captured by the APR-2,
and the LOS data were projected (along the viewing direction) through the APR-2 radar scan to evaluate the cloud structure
along the LOS profile.

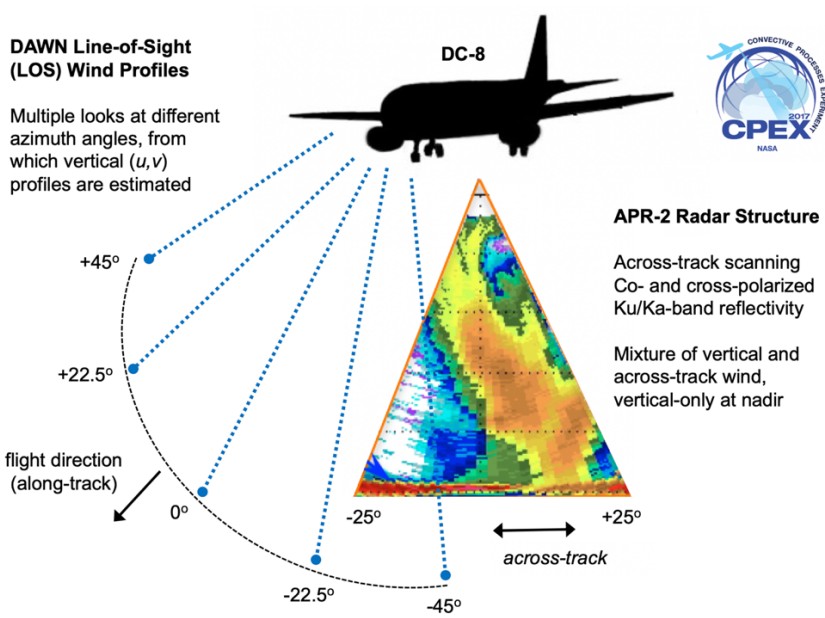

**Figure 1: Depiction of DAWN and APR-2 scanning operations from the DC-8 during CPEX.**



## 3    DC-8 Flight Segments on 10 June 2017.

The intent of this section is to assess the DAWN sampling density near the cloud systems captured by the APR-2, relative to

the cloud evolution.    The 10 June 2017 case is highlighted in this section.    This case is used since it is a fairly isolated cloud growth case, not greatly affected by large-scale forcing at early stages.    On 10 June 2017, the DC-8 took off from Fort Lauderdale near 1800 UTC and headed east towards the area of interest (AOI) with building clouds, located in the box bounded between 24N-26N latitude and 74W-72W longitude.    Figure 2 shows the DC-8 flight tracks on this date, superimposed upon GOES-East geostationary visible channel imagery from 2000 UTC from the JPL CPEX Data Portal

(**http://cpex.jpl.nasa.gov**) (*Hristova-Veleva et. al.*, 2019).

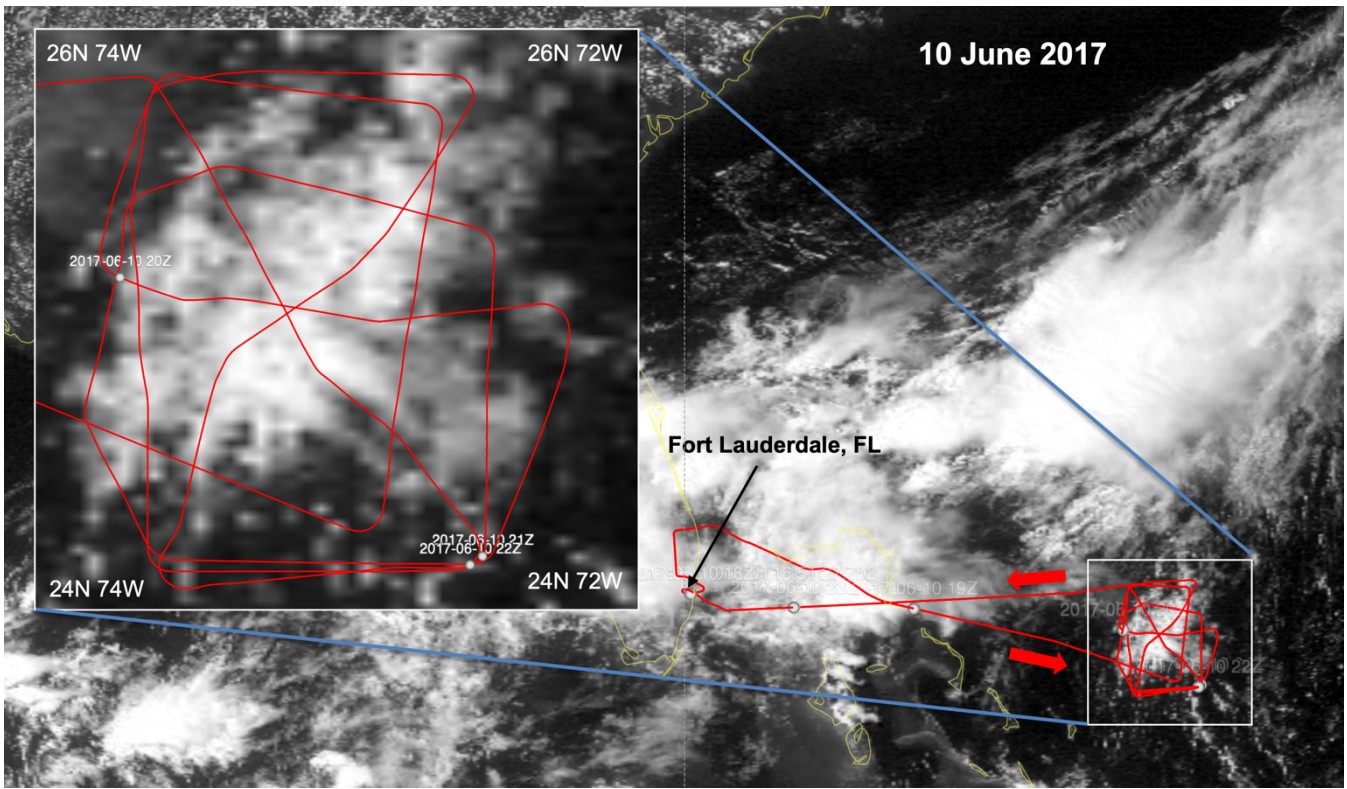

**Figure 2: 10 June 2017 flight track (red lines), shown on the JPL CPEX data portal.  The DC-8 home base at Fort Lauderdale, FL**
**is indicated.    The main area-of-interest is shown in the expanded box, covered by the DC-8 during the 1830-2230 UTC time period.  The grayscale background depicts the GOES-East visible imagery near 2000 UTC.**



A series of convective box patterns were executed, to sample the evolution of the air movement surrounding the convection
from multiple flight bearings. The intent was to be on-station in order to capture developing cumulus clouds before they had
any developed any significant glaciation, before they reached a stage of vertical development where the DC-8 was unable to
overfly from its nominal 10-km flight altitude. A photograph taken from the DC-8 near 2200 UTC (Figure 3) on this date
illustrates an example of a cloud at a desired stage of evolution, where the clouds are captured at an early enough stage such
that the DC-8 can safely overfly multiple times during subsequent evolution.

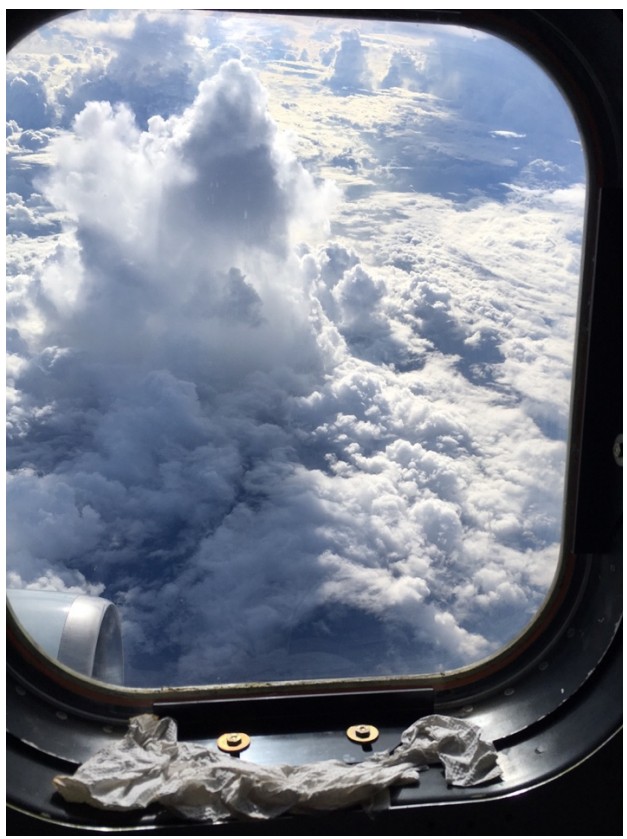


**Figure 3: View of developing cumulus from the DC-8 window, near 2200 UTC on 10 June 2017, from a 10-km flight altitude.**

APR-2 data was collected in tandem with DAWN between 1835-2230 UTC. To explain the DAWN observations relative to
the development of the precipitation, the analysis is broken into four one-hour segments, in each of the sub-sections below.
The APR-2 data will be shown in context to give a sense of when and where (proximity and cloud penetration depth)
DAWN can provide valid wind data. These segments also correspond to the data assimilation interval used in the
investigation of these data by *Zhang et. al.* (2019).



### 3.1    Flight Segment 1 (1830-1930 UTC).

This first DC-8 flight segment collected data outside of the main AOI, whereas the next three flight segments discussed below take place inside of the main AOI.   Figure 4 shows the 1830-1930 UTC plan view at a 2-km (top) and 8-km (bottom) constant elevation levels.   During this time, the DC-8 flew along a 120-degree bearing, heading towards the AOI.   The locations of the DAWN LOS profiles are indicated with colored markers, which owing to the conical scan pattern of the five looks shown in Figure 1, appear as a zig-zag pattern as the DC-8 moves forward (this pattern will be more obvious in

Figures 5 and 6).  Each DAWN LOS beam is colored by the lowest altitude where the SNR > 5 (the 5-dB value is used as a reference level, not as an absolute minimum threshold, as DAWN often provides valid data at lower SNR levels).   During this time the cloud and aerosol conditions were such that the processing of the DAWN LOS data produced a total of 44 wind vectors at 2-km height (top panel), and 70 wind vectors at 8-km height (bottom panel).   The grayscale background indicates the GOES-13 geostationary 11 µm infrared (IR) temperature (white=cold indicating cloud cover, black=warm indicating no

cloud cover) at 1900 UTC for big-picture cloud context.   The locations of these vectors are shown by the red wind barbs. The densest sampling occurs between 1900-1930 in the mostly cloud-free area, shown in the lower right of Figure 4 with 5 m s$^{-1}$ winds at both levels.  There is a tendency for increased directional shear between these levels as the DC-8 approaches the AOI.

**Figure 4. DC-8 flight line during Segment 1 (1830-1930 UTC) on 10 June 2017. The GOES-East IR imagery at 1900 UTC is shown in the background grayscale (white=cold; black=warm). (Top panel): The red barbs show the locations of the wind vectors estimated by DAWN at 2-km height, and the green barbs the profiles from dropsondes at 2-km height. The total vectors are denoted (44 DAWN, 1 dropsonde). The top panel shows the ground locations of each DAWN LOS beam, where the dot color refers to the lowest level where the DAWN SNR exceeded 5-dB. Underlaid over these locations is the average APR-2 Ku-band reflectivity between 2-4 km height. (Bottom panel): Same as top panel, but all DAWN and dropsonde data are for an 8-km height. The APR-2 average Ka-band reflectivity between 7-9 km is shown under each DAWN LOS location. Note the increased sampling density between 1900-1930 UTC.**

To look in more detail to the DAWN sampling proximity relative to the precipitation of individual cloud structures sampled by the APR-2, Figure 5 and Figure 6 show zoom-in depictions covering the two boxes indicated with the orange rectangles in Figure 4, which cover a mostly cloudy area (Box 1 from 1835-1855 UTC, Figure 5) and mostly clear area (Box 2 from 1924-1930 UTC, Figure 6), respectively. The ground locations of the DAWN LOS profiles are indicated with colored markers, and a thin line connected to each marker show the LOS projection from the DC-8. In both Figures 5 and 6, the top panel (2-km level) shows the maximum APR-2 Ku-band reflectivity between 1-3 km plotted underneath the DAWN LOS





locations; in the bottom panel (8-km level) the maximum Ka-band reflectivity between 7-9 km is shown instead (the rationale being that since there is less path attenuation through rain at Ku-band than at Ka-band, the Ku-band data provide a better depiction of the cloud structure for the deeper 2-km level; the APR-2 is more sensitive to clouds at Ka-band than at Ku-band, so the Ka-band reflectivity was used for the higher 8-km level cloud structure). Peak APR-2 Ku-band reflectivities at 2-km exceeded 30 dBZ.

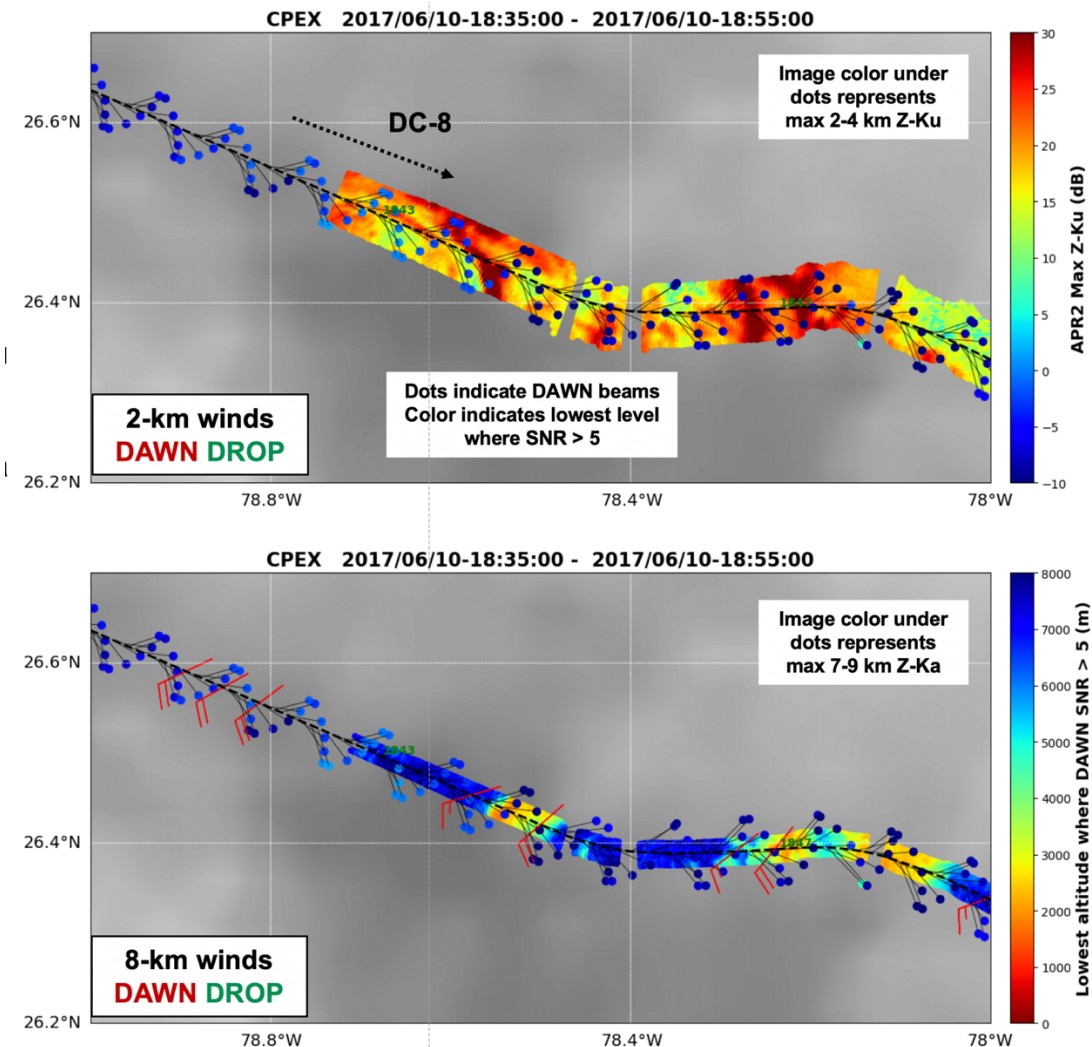

**Figure 5. Same as Figure 4, but zoomed in to the flight segment between 1835-1855 UTC (Box 1 in Figure 4).**

Note that in the mostly-cloudy Box 1 area (Figure 5), the DAWN sampling pattern (depicted in Figure 1) is evident, covering about an 8-km swath as the lidar collects samples at each of the five azimuth locations in its conical scan. For these cloud



cover conditions, no DAWN wind vectors were estimated at the 2-km height. However at 8-km height, DAWN processing

retrieved wind vectors even where the Ka-band reflectivity in the vicinity was as high as about 15 dBZ, showing about 10 m

s$^{-1}$ southeasterly winds.

The mostly-clear Box 2 region shown in Figure 6 (1924-1930 UTC) is presented in an identical layout as Figure 5. At this

time, DAWN was configured in the 2 looks per scan (-45° and 45° azimuth). In this region, DAWN was able to sense well

below 2-km even in the vicinity of clouds at the 10-15 dBZ Ku-band reflectivity level from APR-2, showing 5 m s$^{-1}$

southerly winds at 2-km, becoming more westerly at 8-km height.

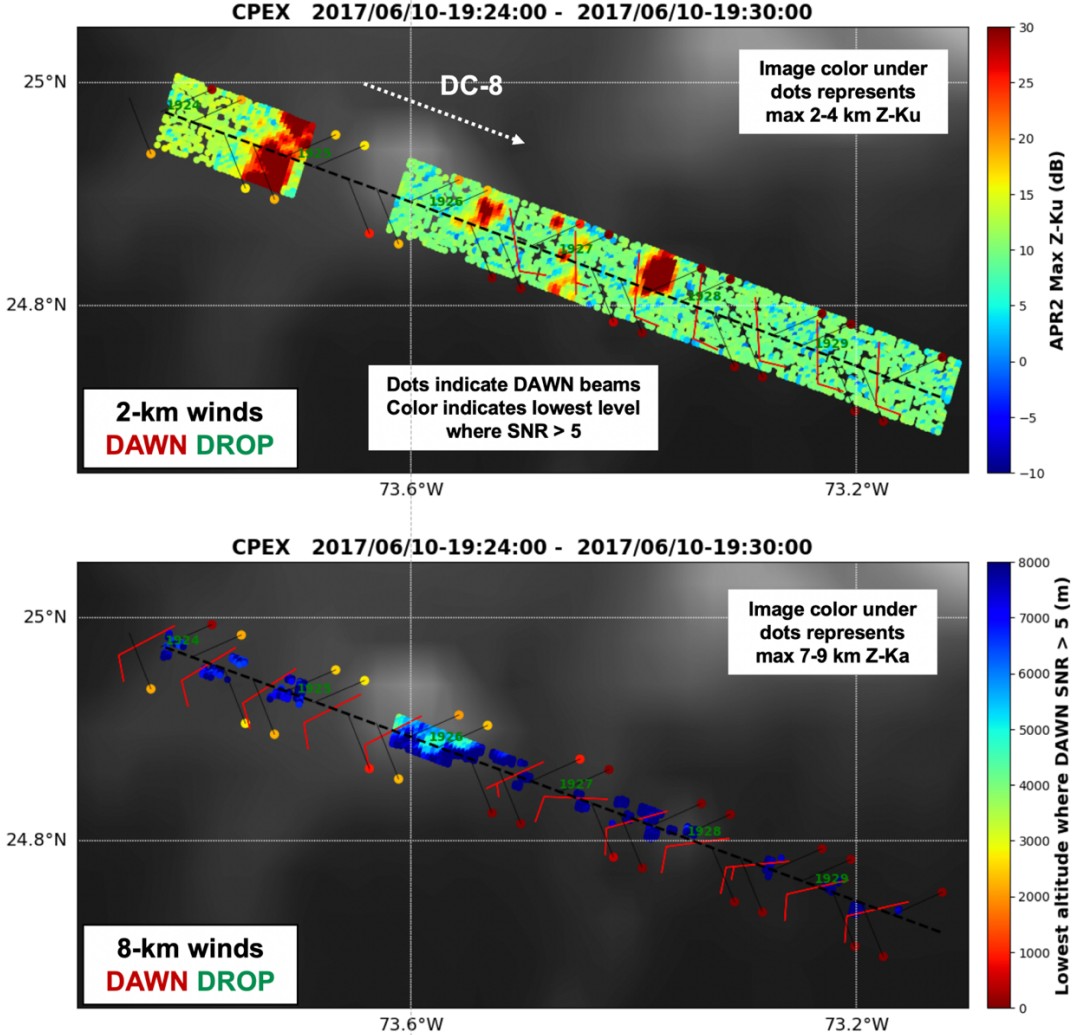

**Figure 6.  Same as Figure 4, but zoomed in to the flight segment between 1924-1930 UTC (Box 2 in Figure 4).**






To provide a depiction of the DAWN vertical sampling capability, a cross-section of the DAWN vertical profile sampling locations superimposed upon the APR-2 nadir reflectivity is shown in Figure 7. The black points represent locations of valid DAWN ($u$, $v$) wind vectors from all DAWN wind profiles during this time. Several notable features are evident. Depending upon the APR-2 transmit pulse length, there is a blind zone (~ 1.8 km) below the aircraft where the radar
processor does not receive any returned signals. DAWN does not have this limitation and provided LOS returns in this missing area. In fact, there is a short period where the cloud tops were within the APR-2 blind zone (near scan 750), but the cloud top was identifiable by the lowest-most level in the DAWN profiles (labeled the "upper cloud area" in gray in Figure 7). Similarly, near the surface where the APR-2 backscatter is affected by ground clutter in the lowest 500-m, DAWN was able to sense to the surface, providing wind observations within the boundary layer. In general, DAWN winds are abundant
above 6-km (where the SNR is highest), and below 3-km (where the aerosol content is higher), with considerable upper level sampling right up to the edges of the tall developed clouds (near scan 1000). There are several DAWN profiles that bump up close to the small convective cell near scan 1800 (denoted with a red ellipse in Figure 7), which are associated with the clouds shown in Figure 6 (Box 2) top panel, where the Ku-band reflectivities exceed 30 dB. To show this area in more detail, Figure 8 zooms in to the Box 2 area (1924-1930 UTC), where three small growing clouds are shown in the middle of
this figure. DAWN wind profiles are produced to the surface next to growing convection near scans 100 and 120, but not for the cell near scan 75. This highlights that convective clouds are not continuous "impenetrable" cloud structures, but in nature have gaps or "holes" in them where the DAWN LOS view can penetrate through to lower levels.



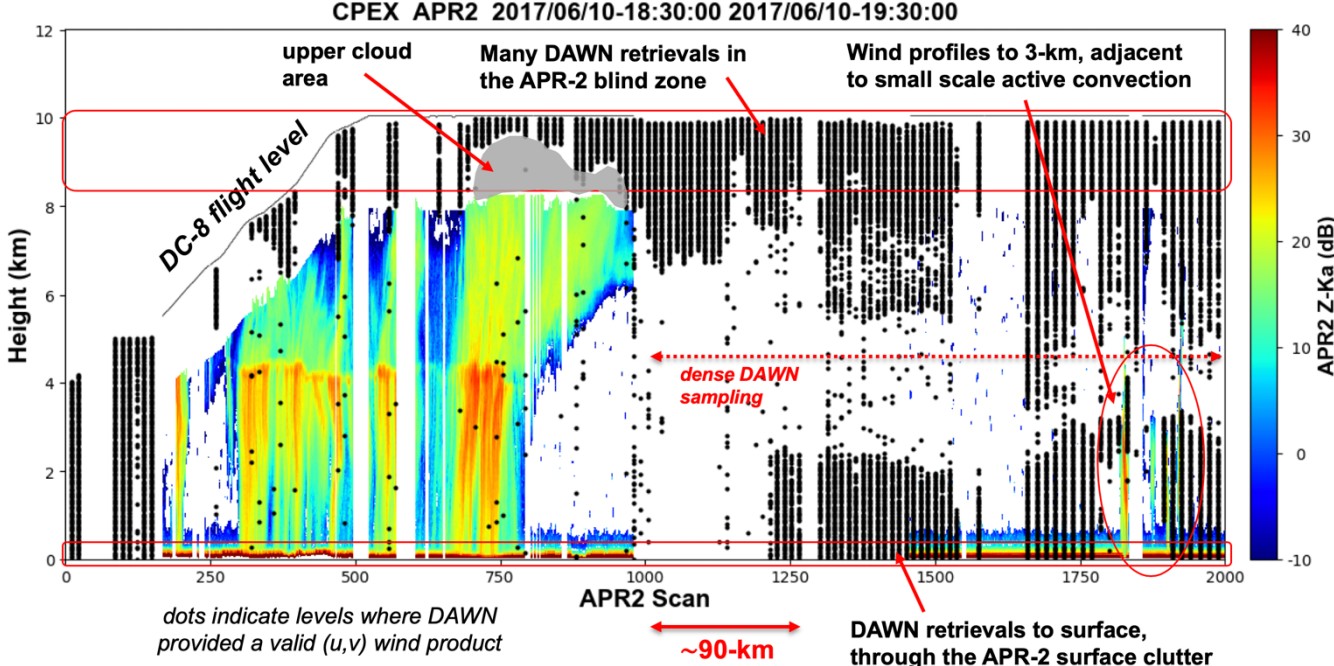

**Figure 7. Cross-section of the APR-2 Ka-band reflectivity (color scale in dB to right) during Segment 1 (1830-1930 UTC). The x-axis represents the APR-2 scan number (2000 scans representing 720-km ground distance), and y-axis the height (km) above the ocean surface. The DC-8 reached its nominal 10-km flight altitude near 1840 UTC. The black points represent vertical locations of valid DAWN (*u, v*) wind vectors from the DAWN wind profiles obtained during processing of the LOS data. The gray "upper cloud" area shows an area where the clouds in the 1.8-km "blind zone" (where APR-2 does not receive any data) but whose cloud top is noted by the lowest-most level in the DAWN profiles.**

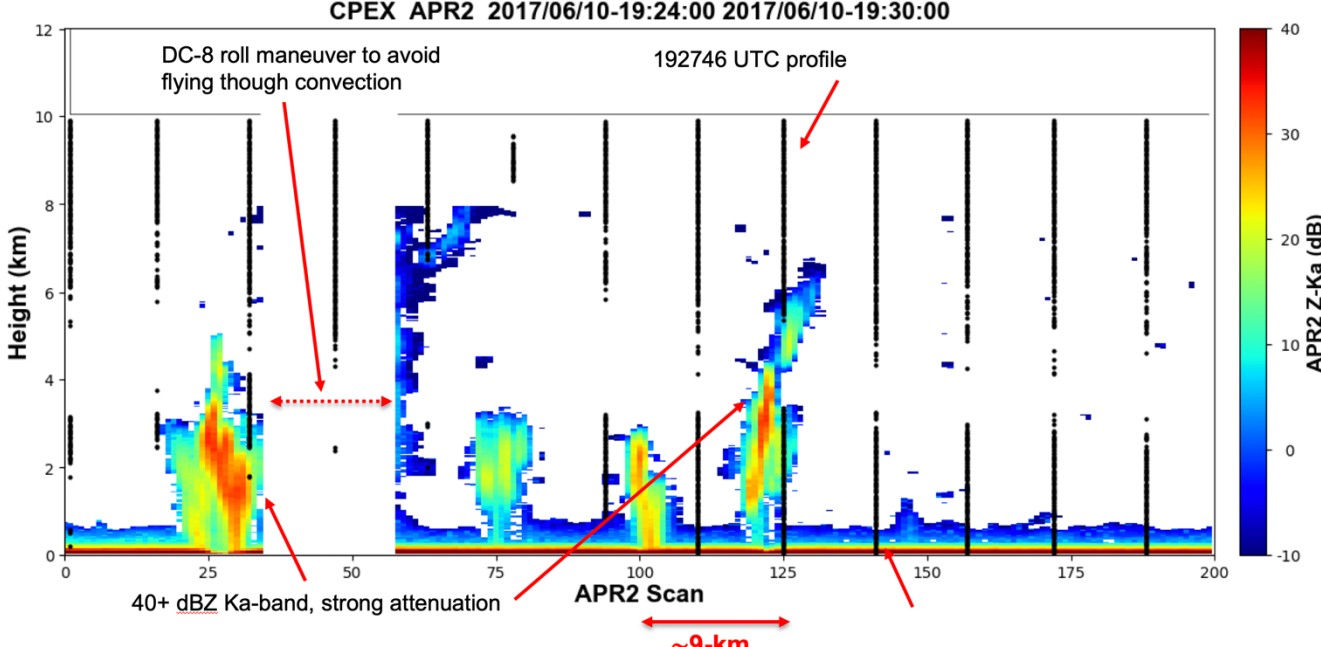

**Figure 8. Same format as Figure 7, but covering only the Box 2 area (1924-1930 UTC) shown in Figure 6. DAWN wind profiles are obtained to the surface very close to the growing convection near 192746 UTC (near scan 120).**


### 3.2    Flight Segment 2 (1930-2030 UTC).

From 1930-2030 UTC, the DC-8 started off along a 120-degree bearing near 25N, then conducted a series of flight legs in a counter-clockwise pattern surrounding clouds at 25N 73W, before leaving to the south and departing along a 270-degree bearing. Figure 9 illustrates the DC-8 flight tracks in the same format as Figure 4, only the 8-km wind level is shown for brevity. Maximum Ka-band reflectivities in the 7-9 km level are near 20-25 dB in the middle of the segment. A total of 38 and 39 DAWN wind vectors were collected at the 2- and 8-km level, respectively, during this time. On the north side of the AOI, the winds were mainly southwesterly near 10 m s$^{-1}$, with 2-km level winds (not shown) more southerly with weaker 5 m s$^{-1}$ speeds. On the south side of the AOI, there was more directional shear, with 8-km westerly winds and 2-km southerly winds (not shown).


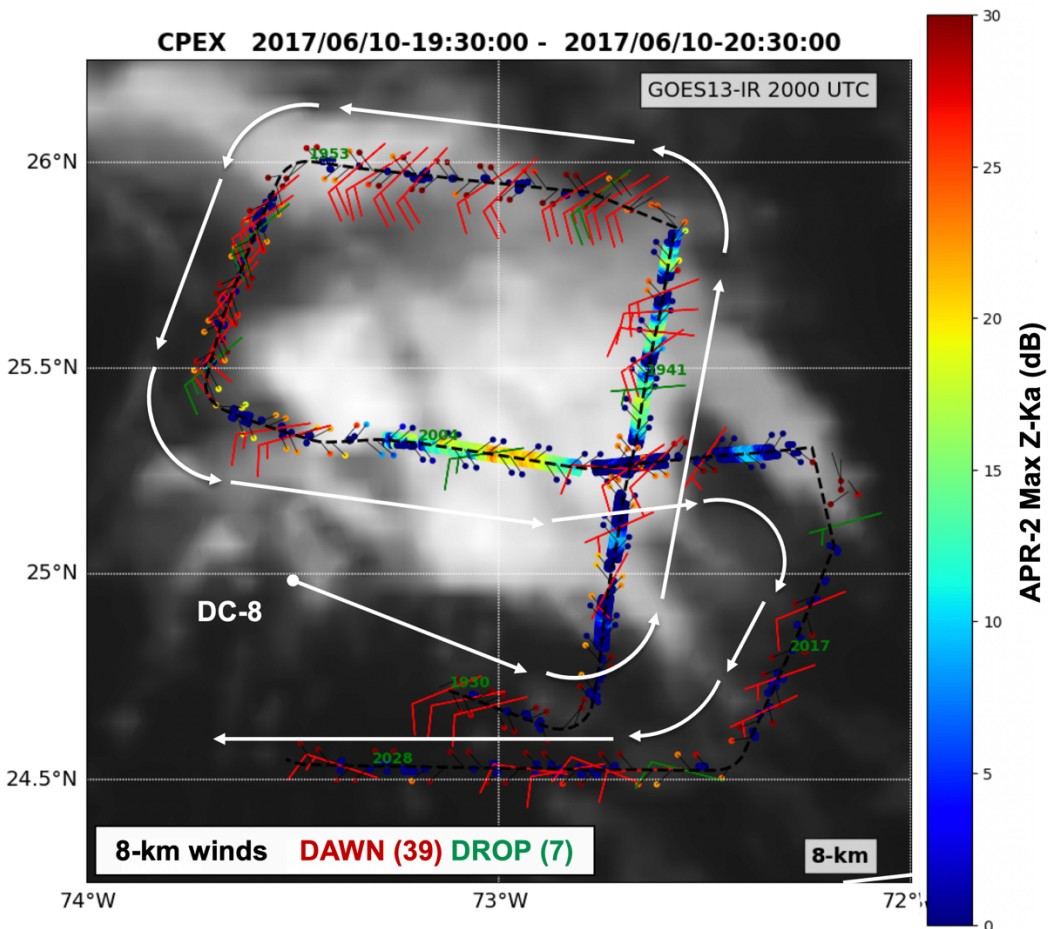

**Figure 9. DC-8 flight line during Segment 2 (1930-2030 UTC) on 10 June 2017. The GOES-East IR imagery at 2000 UTC is shown in the background grayscale (white=cold; black=warm). Same layout and format as the bottom panel (8-km level) of Figure 4.**


The vertical cross section of the of DAWN wind profiles sampling locations alongside the APR-2 nadir reflectivity profile is shown in Figure 10 (areas where the DC-8 was making a banking turn are omitted). Similar to flight segment 1, the two main "no-cloud" regions between APR-2 scans 600-900 and 1300-2000 are well sampled at the upper and lower heights levels. Near scan 850, DAWN data stops near 8-km in areas where APR-2 does not show any cloud, and several profiles

near scan 900 sense deeper (to nearly 4-km), both of which may be from lidar backscatter off of clouds not sensed by APR-2 (i.e., below the minimum Ka-band detectability). The lowest-most level retrieved by DAWN near scan 300 and again near scan 1200 appear to be the cloud top, which occurred in the 1.8-km blind zone (~ 8.2-10 km height) area where APR-2 does not provide any data. Near scan 400, there are numerous DAWN profiles provided in cloud gaps as the DC-8 passed through some higher level clouds.





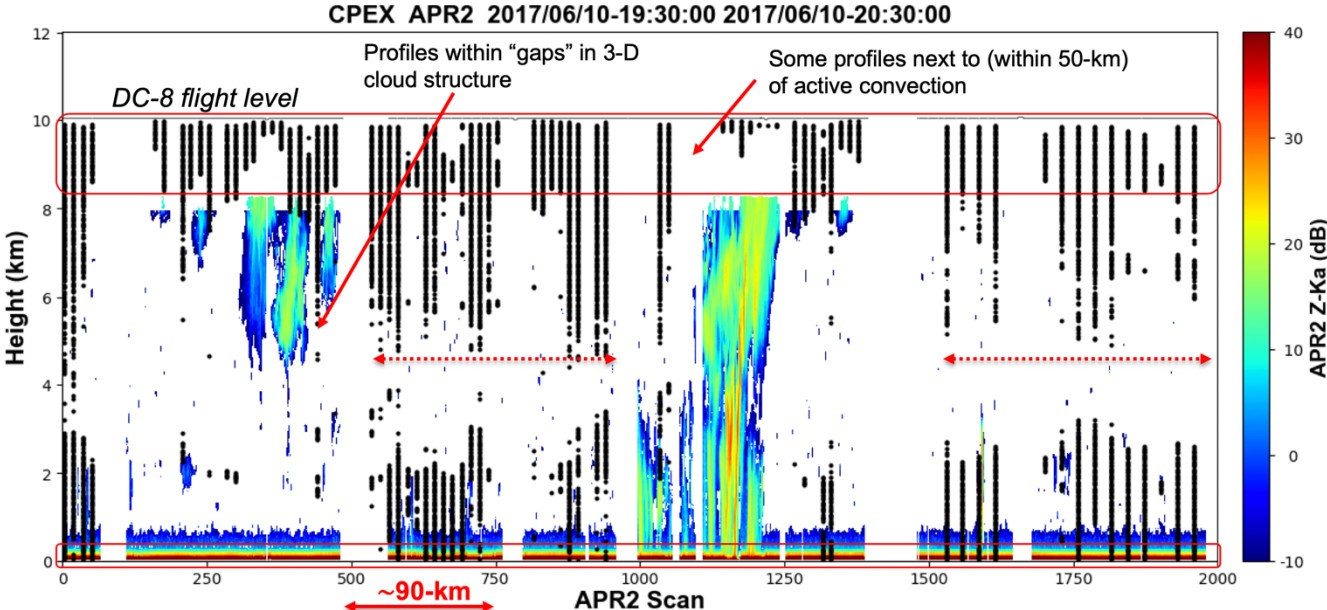


**Figure 10. Cross-section of the APR-2 Ka-band reflectivity (color scale to right) during Segment 2 (1930-2030 UTC). Same layout and format as Figure 7.**

### 3.3    Flight Segment 3 (2030-2130 UTC).

Flight segment 3 begins where flight segment 2 ended, with the DC-8 heading in a northerly direction. The flight revisited some of the area sampled during the previous segment by executing a box pattern in clockwise direction, before exiting to the east along a 90-degree bearing (Figure 11). Towards the end of this flight segment, the DC-8 dropped to a 9-km flight level. At 8-km height, 25 DAWN wind vectors were estimated from the LOS data (at 2-km, 18 DAWN vectors were available). The weak directional shear on the south and southwestern regions of the AOI is still present, with 8-km westerly

winds and 2-km southerly winds (not shown) near 5 m s$^{-1}$ speeds. On the north side of the AOI, 8-km winds are mostly southwesterly near 15 m s$^{-1}$. At the 2-km level, 18 DAWN wind vectors were estimated, nearly all concentrated on the south side of the AOI, but several southeasterly 5 m s$^{-1}$ winds were estimated on the north side of the AOI. Figure 12 shows the DAWN vertical sampling density during this flight segment relative to the APR-2 Ka-band reflectivity structure. On the east and south sides of the AOI the DC-8 passed above a region of thin clouds (as shown in the IR background in Figure 11), but

were not detected by (i.e, below the sensitivity of) APR-2 except for some 2-km cloud tops near scan 1100. Overall in this region, DAWN sampling was reduced in the 2-8 km height level, but the E-W leg (scans 1200-1400) provided profiling to the surface in many locations.


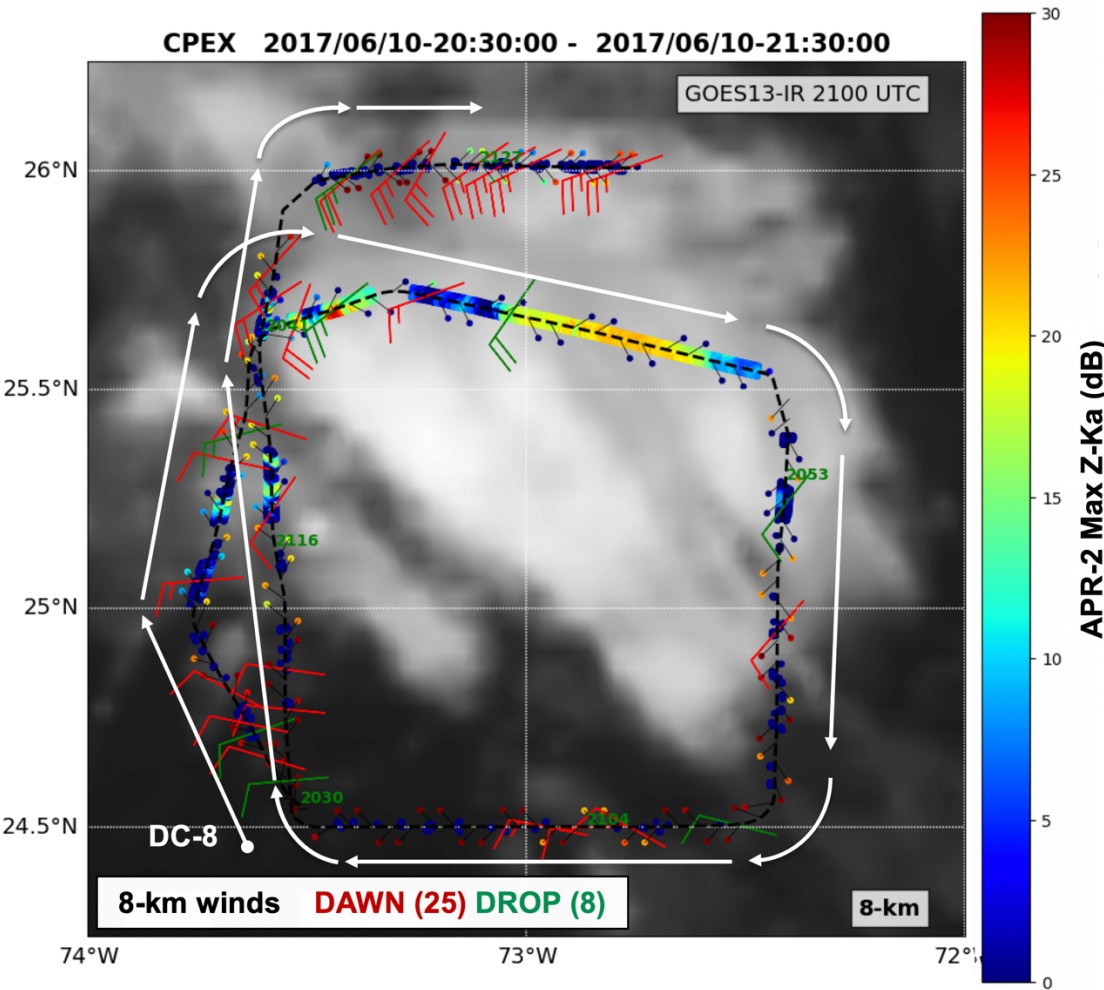

**Figure 11.** DC-8 flight line during Segment 3 (2030-2130 UTC) on 10 June 2017. The GOES-East IR imagery at 2100 UTC is shown in the background grayscale (white=cold; black=warm). Same layout and format as the bottom panel (8-km level) of Figure 4.

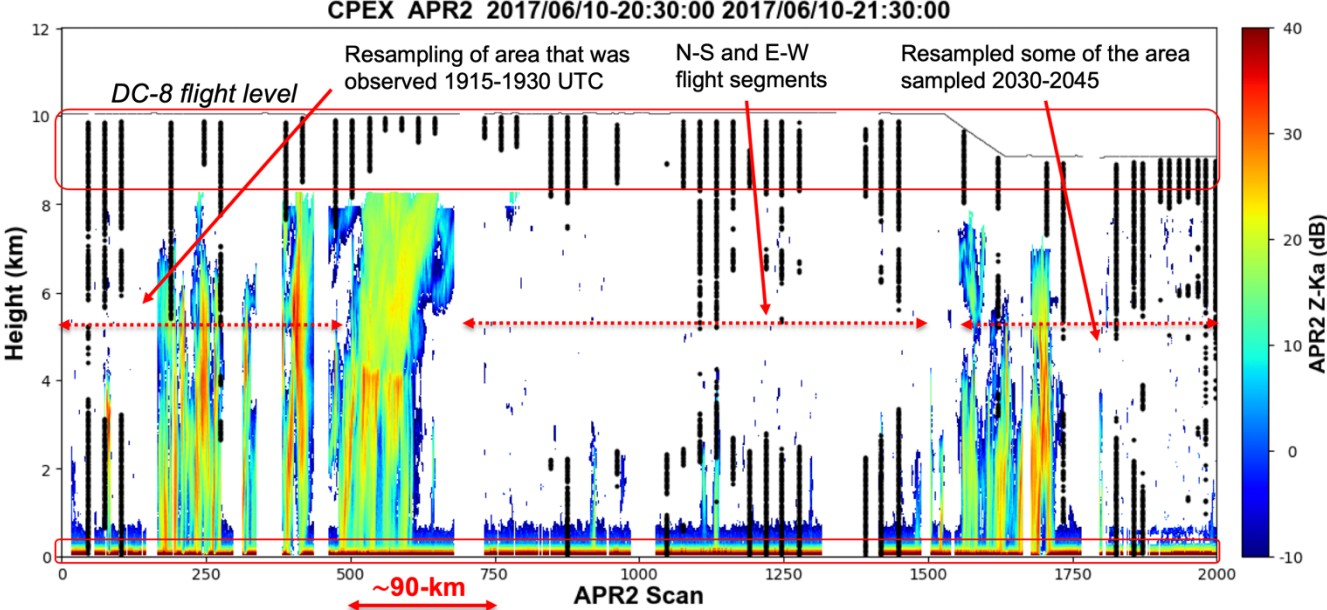

**Figure 12. Cross-section of the APR-2 Ka-band reflectivity (color scale to right) during Segment 3 (2030-2130 UTC). Same layout and format as Figure 7.**

### 3.4 Flight Segment 4 (2130-2230 UTC).

Flight segment 4 begins with the DC-8 heading in an easterly direction and then banking to a 225-degree bearing. The DC-8 partially completed a figure-eight pattern, before exiting to the west along a 270-degree bearing and returning to Florida, as shown in Figure 13. The overall DAWN profile sampling numbers are higher than segment 3, with 49 and 63 DAWN vectors provided at 2- and 8-km heights, respectively. Small growing clouds were first overflown during scans 200-400. The cloud system near 25.5N 73.5W has matured considerably relative to its structure in previous flight segments, with a fairly well-defined bright band shown near scans 1450-1550.

DAWN vertical sampling density during this time is fairly dense (Figure 14), with more winds provided in the 2-6 km height level than during flight segment 3, notably in the middle and end of this flight segment. When the DC-8 moved to a lower 9-km flight level, the pulse width was changed resulting in the APR-2 blind zone being shorted by one-half (to 0.9 km), which is evident for the tallest clouds near scans 400 and 1400. DAWN also provided overall better sampling in the mid-levels from this lower flight altitude, with almost complete top-bottom profiles towards the end of the flight segment.





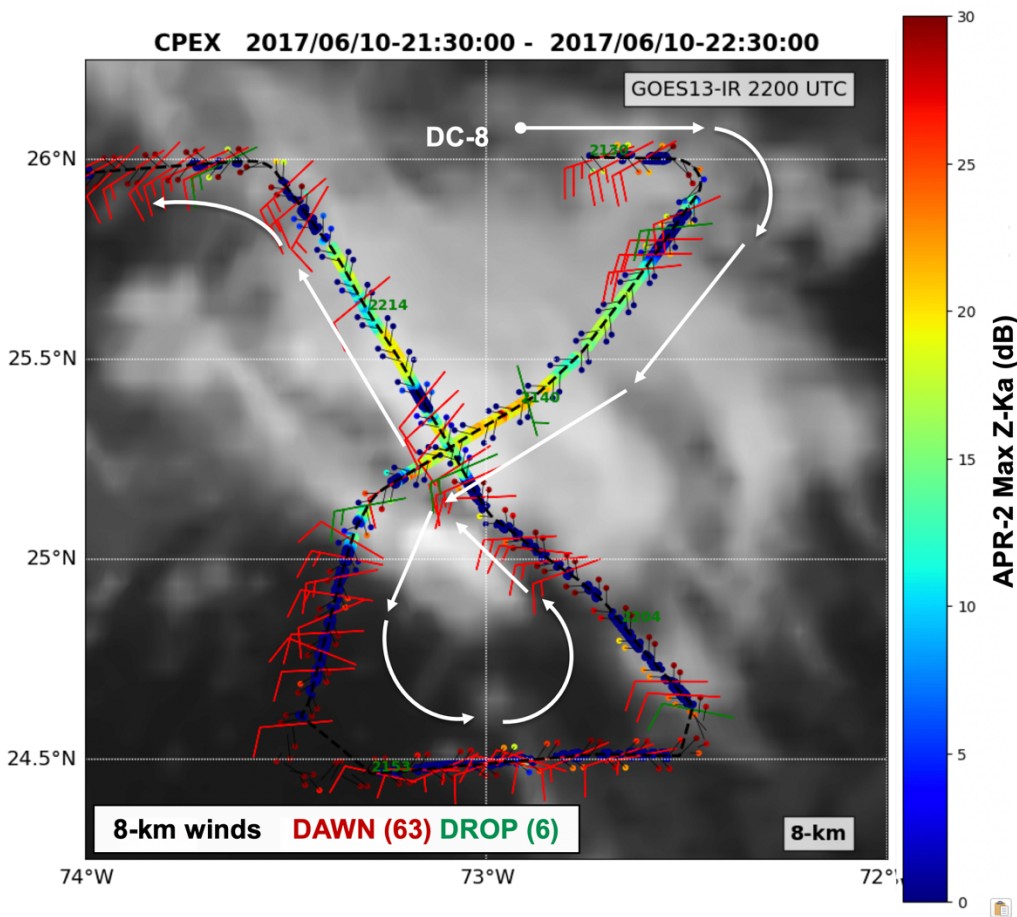

**Figure 13.** DC-8 flight line during Segment 4 (2130-2230 UTC) on 10 June 2017. The GOES-East IR imagery at 2200 UTC is shown in the background grayscale (white=cold; black=warm). Same layout and format as the bottom panel (8-km level) of Figure 4.



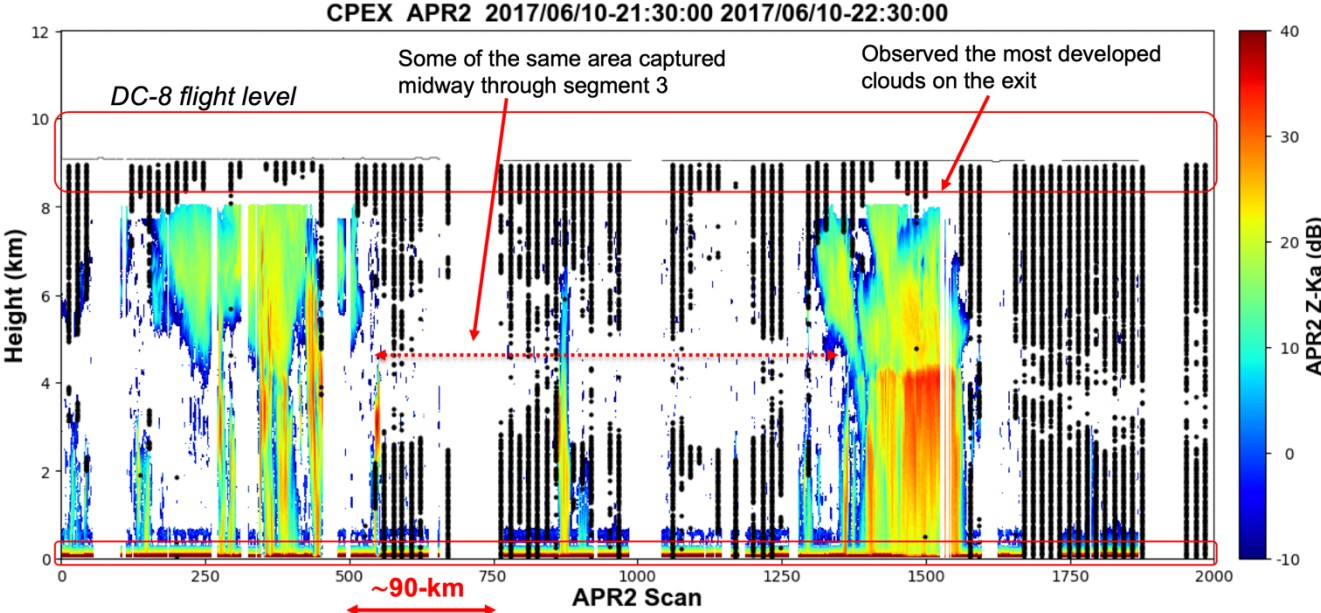

**Figure 14.** **Cross-section of the APR-2 Ka-band reflectivity (color scale to right) during Segment 4 (2130-2230 UTC).** Same layout and format as Figure 7.

## 4        DAWN and APR-2 horizontal winds on 11 June 2017.

APR-2 also provided vertical air motion and structure of the cloud systems in the cloud-detected regions where DAWN's profiling capability was degraded. The purpose of this section is to examine a method to couple the two wind estimates near clouds. By viewing clouds from multiple viewing directions near nadir, airborne Doppler radars sample a mixture of the vertical and horizontal winds associated with the movement of the hydrometeors being sensed (*Heymsfield et. al.*, 1996). As the DC-8 moves forward and the APR-2 scans across-track, the measured Doppler velocity represents a combination of the vertical and across-track components of the hydrometeor motion within each APR-2 range bin (*Durden et. al.*, 2003). These data can provide some complementary wind direction information to complement DAWN, and under the right conditions (no significant horizontal shear across the APR-2 scan swath) provide some continuity in the wind measurements between the cloud and no-cloud areas. The received Doppler velocity represents contributions from the motion of the hydrometeors owing to air motion, and the contribution owing to the (reflectivity-weighted) hydrometeor fall speed. Define $\theta$ as the viewing angle from nadir (e.g., zero represents straight downward, and negative and positive denote the left and right sides of the APR-2 swath, respectively), and $v_z$ and $v_y$ as the vertical and across-track wind components. Then the Doppler wind at corresponding left and right sides of the swath is given by:

$$v_{left} = v_z cos|\theta| - v_y sin|\theta| \qquad\qquad (1)$$



$$v_{right} = v_z cos|\theta| + v_y sin|\theta| \qquad (2)$$

where the subscripts left and right refer to the corresponding APR-2 beam positions at $-\theta$ (left side of swath) and $+\theta$ (right side of swath), respectively. The vertical (z) and across-track (y) wind components are easily solved for,

$$v_z = (v_{right} + v_{left})/2cos|\theta| \qquad (3)$$

$$v_y = (v_{right} - v_{left})/2sin|\theta| \qquad (4)$$

Note that in this formulation, the effects owing to the hydrometeor fall speeds are still included, so the estimate of $v_z$ in (3) is
not the same as the vertical ($w$ component) wind due to air motion only. To account for the fall speed, the fall speed-reflectively relation developed by Black et al. (1996) is applied and only the 8-km level winds (where there has not yet been significant attenuation) are assessed. After this correction, $v_z$ is assumed equal to the $w$ wind owing to air motion. However, in general more rigorous radar inversion methods that account for the radar attenuation and the hydrometeor Doppler fall speed are required before this formulation can be applied to lower cloud levels (*Guimond et. al.*, 2014)


This principle is examined on the APR-2 data gathered between 1800-2100 on 11 June 2017. Figure 15 shows the plan view of the 8-km level winds from DAWN, providing abundant winds, including some vectors that are close to clouds, as shown by the APR-2 Ka-band reflectivity at this level. The DC-8 entered the area from the northeast. There are six flight legs along a predominant 90-degree (W-E) or 270-degree (E-W) (+$u$ and -$u$ wind component direction, respectively) flight
bearings, beginning near 1800, 1815, 1838, 1900, 1920 and 1955 UTC, with some slight deviations along these directions to avoid deep clouds near flight level. The first and last three of these flight legs occurred in predominantly cloud-free and cloud-covered conditions, respectively. The top panel of Figure 16 shows the time intervals corresponding to these 90- and 270-degree bearings. In these flight directions, the APR-2 across-track wind component (4) is contributed by the $v$ wind. When the DC-8 transitions from a 90- to 270-degree flight bearing (or vice-versa), a flip in the sign of the APR-2 $v$
component is expected, since the APR-2 right swath side becomes the left swath side.


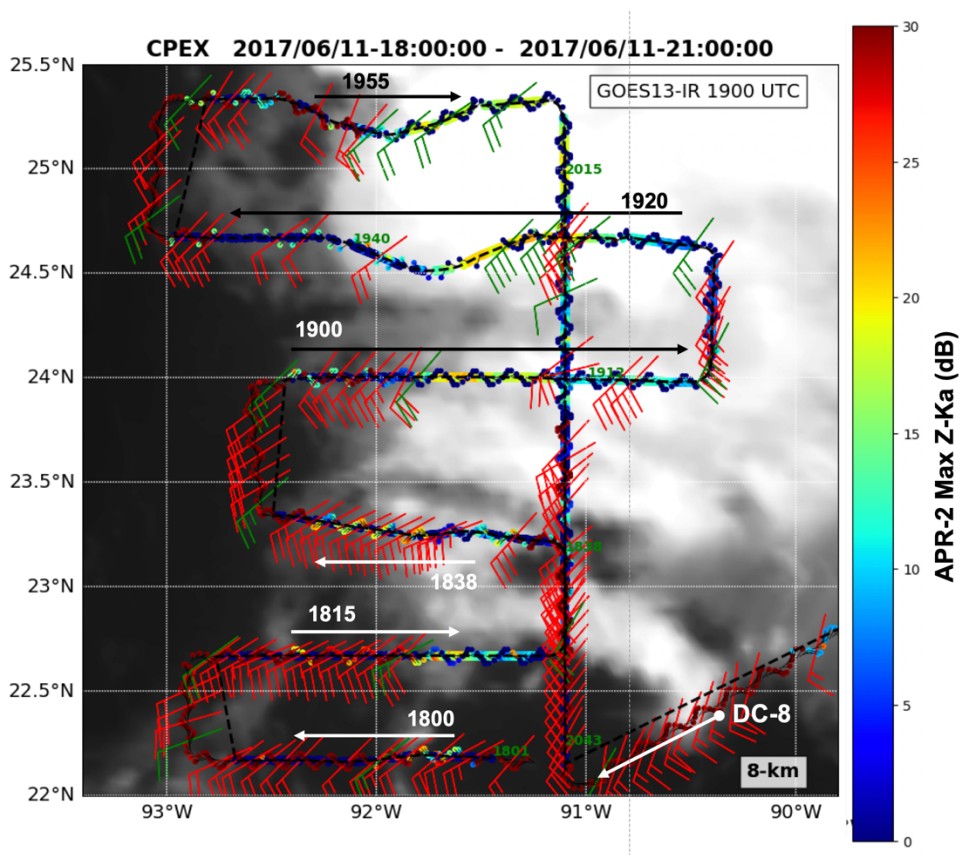

**Figure 15. DC-8 flight line during 1800-2130 UTC on 11 June 2017. The GOES-East IR imagery at 1900 UTC is shown in the background grayscale (white=cold; black=warm). Same layout and format as the bottom panel (8-km level) of Figure 4. The six E-W or W-E flight lines beginning near 1800, 1815, 1838, 1900, 1920 and 1955 UTC are shown.**

The bottom panel of Figure 16 shows the APR-2 vertical ($w$) and across-track ($v$) winds estimated from (3) and (4), plotted in orange and black colors, respectively. DAWN ($u, v$) winds at the same 8-km level are shown in red and blue colors, respectively. Near 1830 UTC, the DAWN $v$ component is near 5 m s$^{-1}$, and the APR-2 $v$ component is near 5-10 m s$^{-1}$, but quickly (within a few minutes) changes to a smaller value as the DC-8 enters an area with stronger vertical motion and assumptions on horizontal shear are likely voided. Near 1840 when the DC-8 is flying along a 270-degree bearing and detects clouds at the 8-km level, the APR-2 $v$ component changes to -12 m s$^{-1}$. While it is the expected wind speed sign flip, it is more difficult to compare the wind speed magnitude. Also, the 270-degree bearing has some deviations near 1843 UTC to avoid convection at flight level.



**Figure 16. (Top)** DC-8 heading between 1800-2130 UTC on 11 June 2017, highlighting the six time periods depicted in Figure 15.
**(Below)** DAWN (*u, v*) wind vectors at the 8-km level (red and blue points, respectively). APR-2 (*w, v*) winds (orange and black, respectively) estimated from (3) and (4).

A second coincidence occurs between the APR-2 data near 1910 and 1925 UTC, where the APR-2 v component flips sign between similar wind speed values. However, the area at 1925 UTC is so cloud-filled that there are no nearby DAWN wind
profile data to compare to. It also represents an area with stronger vertical winds, where the assumption of no significant horizontal shear across the APR-2 scan swath is likely not valid. While this is not a rigorous comparison of DWL and Doppler precipitation radar horizontal winds, the principle could be applied to any arbitrary pair of DC-8 flight bearing segments that are separated by 180-degrees. The winds estimated by (4) are more generally a combination of (*u, v*), and the DAWN (*u, v*) winds could be transformed to these same directions for comparison. This complement of Doppler radar and
DWL observations could provide a means to link horizontal wind data outside of clouds and inside clouds (away from strong vertical motion, from APR-2), an important transition region. Space-based Doppler radar measurement methods to estimate the horizontal LOS (HLOS) wind in-cloud have been proposed (*Illingworth et. al*., 2018), as one means to complement the



HLOS winds from Aeolus. However, further investigation from CPEX and other APR-2 airborne data are needed to assess the quality of the radar wind components before they can be used for science or model data assimilation purposes.

**5        Conclusions.**

This manuscript has presented joint observations from the DAWN Doppler wind lidar and the APR-2 (Ku/Ka-band) Doppler precipitation radar, collected during the CPEX campaign in 2017. Data from NASA DC-8 flight segments from two flight dates were examined to examine the ability of DAWN to sense air motion nearby to developing convection.       The flight patterns on June 10-11 were selected for this purpose. For the June 10 flight date, the DC-8 arrived on-station to the area of

interest, with sufficient time to capture the evolution of isolated, small-scale (< 10-km horizontal extent, many not yet glaciated) clouds from numerous DC-8 repeat passes for about a 3 hour period.  The environment surrounding the clouds on this date exhibited relatively weak directional shear between the 2- and 8-km levels in the area south and southeast of the developing convection.  A number of growing convective clouds with APR-2 echo tops below 5-km were sampled by the APR-2, away from the more developed convection.  The capability of DAWN to collect LOS profiles near convection was

highlighted for several passes where profile retrievals were possible up to the edges of many APR-2 detected cloud systems. On June 11, the DC-8 sampling pattern consisted of successive repeat passes on E-W and W-E flight bearings, where the cross-track winds from APR-2 were examined for consistency with nearby DAWN winds, in the proximity of cloud edges.

As stated in the introduction, this manuscript provides the observational context for a separate mesoscale model data

assimilation study, which is aimed at quantifying the impact of the DAWN measurements on the analyzed atmospheric state variables and on the forecasted precipitation when the DAWN wind profile observations were assimilated into the model (*Zhang et. al.*, 2019).  While only limited examples are shown, these particular findings highlight the importance of when and where the wind observations are taken, and will aid in assessing future requirements and limitations on the scale (horizontal and vertical) of the observations needed for future airborne field campaigns with similar instrumentation.

**Data Availability**

The DAWN LOS and profile data (ASCII text format) and APR-2 data (HDF5 format) are available from any of the JPL authors upon request. Data volumes for DAWN and APR-2 are approximately 100 MB and 1 GB per CPEX flight date, respectively.



**Team list**

**Author Contribution**

ST, SLD and OS carried out the APR-2 data pre-processing to produce Level-1 reflectivity products. FJT carried out the data alignment between DAWN and APR-2. All JPL authors contributed to operations of the APR-2 during CPEX. SG and DE collected and performed all DAWN data processing.

**Competing Interests**

The authors declare that they have no conflict of interest.

**Acknowledgements**

The work contained in this presentation was carried out at the Jet Propulsion Laboratory, California Institute of Technology, under a contract with NASA. © 2020 all rights reserved. Support from NASA under the Weather and Atmospheric Dynamics program is recognized. The authors gratefully acknowledge the DC-8 flight support team, and the CPEX Co-investigators Dr. Ed Zipser and Dr. Shuyi Chen.

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
