# Peer review of "Joint Analysis of Convective Structure from the APR-2 Precipitation Radar and the DAWN Doppler Wind Lidar During the 2017 Convective Processes Experiment (CPEX)"

_Atmospheric Measurement Techniques, 2020_

## Referee Comment (RC1) · Anonymous Referee #1 · 17 Apr 2020

The paper presents some very interesting measurements collected during the CPEX experiment by joint Doppler Aerosol Wind Lidar and a dual frequency Doppler radar observations. The paper is very important because lays the foundation on how to integrate these two different Doppler observing systems. The paper is generally very well written. I am looking forward seeing the data used for a better understanding of the linking between 3-D air motion and cloud structure in a peer-reviewed journal.

I have mainly some comments to improve the layout and to add the information content of some of the figures. Also Sect4. could be improved.

[Figure]

Line 140: "any developed " ==> developed

Fig.4: it is very difficult to read this figure. In particular the overlapping of the image colour and the coloured dots is particularly troublesome. Why not shifting the dots upwards by 0.5 degree latitude (properly commenting on that in the caption) ?

Fig5: maybe it is worth saying that no image colour is present if no clouds with reflectivity above radar sensitivity are present in the layer

Fig7: colour-scale is in dBZ not dB, right? (also line 223 and through the document)

Fig8: red box: If the red box represents the blind zone it should follow the aircraft flight level and go oblique before scan 500. "above 6-km (where the SNR is highest), and below 3-km (where the aerosol content is higher)" it is a little bit misleading because I think in both cases the SNR is high, in the first case because of the shorter range, in the second for the higher backscattering. In general it is not clear to me why between 1000 and 1500 (there is not a clear range dependence in the upper part, is the lower part structure related to aerosol in the first two km?) the black dots are distributed like they are. Maybe over-plotting lidar SNR contour levels could help. Same applies to Fig.10-12-14. Also isn't in all such figures a lost opportunities? Why not showing for some of the black dots the wind direction? We could actually appreciate wind shear in proximity of convective clouds.

Fig.8: about the "continuous "impenetrable" cloud structures" comment obviously the lidar will see through the 3D structure, no question. I am a little bit sceptical about the profile at 192746 UTC; I cannot imagine that the lidar signals goes through the black dots as currently drawn; are we guessing here that there is basically no cloud liquid for that specific path and light will go through rain and ice? otherwise couldn't we argue that the path maybe a little bit different from the one currently drawn (you have pointing uncertainties to account for, haven't you?)?

Sect.4: I understand that the retrieval of wind must be done in the aircraft reference of

frame but for the interpretation it is much better to go back to the usual system (E-W and N-S winds). Since the DC-8 heading is known this is a simple conversion. By so doing you will get rid of all the discussion about the heading and we will actually see the "real winds" (which are the relevant ones for the study of "dynamical processes"). Also the u,v notation is confusing since it is typically used for E-W and N-S winds.

---

## Referee Comment (RC2) · Anonymous Referee #2 · 22 Apr 2020

General comments

This well written manuscript presents novel, unique and relevant collocated airborne Doppler lidar and radar measurements in complex, convective subtropical environments. Focus is set to a common display of both instruments' data sets from two exemplary NASA DC-8 research flights, to show regions of common data overlap and the measurement limits of each instrument. The paper is fully suitable to the scope of AMT, and both the scientific relevance and the data quality are outstanding. However, or because of this, I find it disappointing that scientific conclusions from such an interesting instrument combination are missing, and that the reader is just referred to future studies.

There exists a WCRP Grand Challenge on Clouds, Circulation and Climate Sensitivity (which the authors do not address - anyway), because one of the big science questions is the feedback of convection on dynamics. This DC-8 instrumentation is perfectly suited to address this question, and I think the readers would like to see more details and (preliminary) conclusions hereto.

So here is my suggestion how to avoid disappointment without too much extra effort. The interesting questions are: what is the evolution in time of the probed cloud cluster? Did you observe secondary circulations due to cloud growth? Is convergence or divergence visible in the measurements? So, at the end of section 3, maybe also 4, you should answer these questions. An extra, perhaps 3-d, sketch of the cloud cluster with the essential wind arrows resulting from all flight segments at both 2- and 8-km heights would be very helpful.

Specific comments

The abstract is misleading. It repeats the nicely written overarching science issues from the introduction (it is OK to address them in the introduction), suggesting to the reader that these big questions are the main topic (which is not quite correct), but in fact it lacks the major results of the paper (= the answers to my above questions). Furthermore, it mentions "transport of water vapor" (line 15) and "Frequent dropsonde data" (l. 19) which both are not major topics of the paper.

Lines 37 and 64: these sentences highlight the importance of the vertical distribution of water vapor, a very interesting topic, yet which is (unfortunately) not addressed in this paper. The dropsondes could provide the humidity profiles, but I guess there were too few of them on these two flights to make solid statements, and/or this topic is beyond the scope of this study? Could you comment on this?

Line 79: ". . .radar and DWL observations from two exemplary flight days" to be more precise. In this context, I find that Table 1 is not at all needed.

Fig 1 is difficult to understand. Fig 5 suggests that at 2 km asl the +-45° lidar positions at 30° off-nadir angle have about the same separation than the APR swath width at +-25°. Could that be illustrated in Fig 1?

Line 125: please explain the synoptic situation, and in more detail why you chose this particular situation out of 16 flights. Is the isolated cloud cluster you probed a beginning MCS? What is its relation to the extended cloud band to the north? Was this situation typical for the whole campaign, or was it a very particular "golden day"?

Fig 4 is very difficult to understand: what if you would swap Fig 4 and Fig 5? Beginning with the zoom, it would be much easier to understand the heavy full segment 1 overview. Why is the dropsonde at different places? Were there two different dropsondes? Please explain.

Line 225 and caption of Fig 8: explain why you think the convection is growing. Fig 15: is green still the dropsondes? So there are many more dropsondes on this day? Please explain.

Technical corrections

Line 45: 2x "associated", and verb is missing.

Line 107: remove "highly capable".

Line 110: " a constant 30° elevation angle", I do not understand, you mean probably an off-nadir angle of 30°.

Fig 2: the GOES image in the expanded box is very coarse, it should be available at much higher resolution if this is a visible imagery, and lat/lon indications would make the big image easier to interpret. Also, highlight the 4 segments from sections 3.1 - 3.4.

Line 140: "before they had developed.."

Lines 160 and 176: the unit dB is probably wrong when characterizing a DAWN SNR level.

Line 185: "show the LOS projections to msl from the DC-8"

Fig 5: in the upper panel there are no winds, and in the lower panel there is no drop-sonde, so you may want to adapt both lower left text boxes.

Fig 6: I do not see any dropsonde, so you may want to adapt both lower left text boxes.

Line 249: "winds (not shown)", you could refer to Fig 6, showing a region quite close where the winds are shown.

Line 256: "of the DAWN..."

Figs 9, 11, 13 and 15: the color bar is too large.

Line 393: 2x "examine"

---

## Author Comment (AC1) · 3 May 2020

The paper presents some very interesting measurements collected during the CPEX experiment by joint Doppler Aerosol Wind Lidar and dual frequency Doppler radar observations. The paper is very important because lays the foundation on how to integrate these two different Doppler observing systems. The paper is generally very well written. I am looking forward seeing the data used for a better understanding of the linking between 3-D air motion and cloud structure in a peer-reviewed journal.

I have mainly some comments to improve the layout and to add the information content of some of the figures. Also, Sect 4 could be improved.

Line 140: "any developed" ==> developed

Fixed.

Fig.4: it is very difficult to read this figure. In particular the overlapping of the image colour and the coloured dots is particularly troublesome. Why not shifting the dots upwards by 0.5 degree latitude (properly commenting on that in the caption) ?

We agree, in fact since Fig 5 is the figure that is supposed to represent the DAWN LOS sampling, we just removed these dots from this figure and the color scales represents the Ku and Ka-band reflectivity. In Figures 6 and 7, which zooms in to the various LOS profiles, the second colorbar for the lowest level is reinstated.

Fig5: Maybe it is worth saying that no image colour is present if no clouds with reflectivity above radar sensitivity are present in the layer.

We have added this (no image color= no clouds present that are above the APR-2 radar sensitivity).

Fig7: colour-scale is in dBZ not dB, right? (also line 223 and through the document)

Yes, that's correct. We have fixed this terminology throughout.

Fig8: red box: If the red box represents the blind zone it should follow the aircraft flight level and go oblique before scan 500. "above 6-km (where the SNR is highest), and below 3-km (where the aerosol content is higher)" it is a little bit misleading because I think in both cases the SNR is high, in the first case because of the shorter range, in the second for the higher backscattering. In general, it is not clear to me why between 1000 and 1500 (there is not a clear range dependence in the upper part, is the lower part structure related to aerosol in the first two km?) the black dots are distributed like they are. Maybe over-plotting lidar SNR contour levels could help. Same applies to Fig.10-12-14. Also isn't in all such figures a lost opportunity? Why not showing for some of the black dots the wind direction? We could actually appreciate wind shear in proximity of convective clouds.

One reason for the "lower part structure" referred to is likely due to clouds that may be present, but below the APR-2 detectability, so we have no "proof" that they are really there. Some of these clouds are "thin" enough that DAWN can penetrate and still have sufficient dynamic range (see the reply to the comment below with an example figure of this situation), others penetrate only partially. Or some profiles occurred during slightly different aerosol concentration in the lower 1-2 km than did other nearby profiles.

The 2-km and 8-km wind barbs are plotted on Figure 4, which corresponds with Figure 7. In other words the flight segment shown in Figure 4 maps one-to-one with the x-axis on Figure 7. If I add anything more to Figure 7, it will clutter it up.

But your comment is a very good one which gave me an idea to present the wind hodographs for each of the four one-hour time segments. From these, the directional wind shear (if present) is more obvious. These are now included in the paper. I did this for the 2-vs-8 km levels and the 2-vs-6 km levels (Figure 7, similarly for other segments). Which show very interesting shear, especially in the 1830-1930 period. Furthermore, I separated the hodographs into quadrants (NE, SE, SW, NW) relative to the approximate center (25.2N 73W) of the flight box on this date. This shows the sustained 2-8 km shear in the area SW of the area of interest, that flips sign by about 90-deg when compared to the 2-6 km shear.

[Figure]

Fig.8: about the "continuous "impenetrable" cloud structures" comment obviously the lidar will see through the 3D structure, no question. I am a little bit skeptical about the profile at 192746 UTC; I cannot imagine that the lidar signals goes through the black dots as currently drawn; are we guessing here that there is basically no cloud liquid for that specific path and light will go through rain and ice? otherwise couldn't we argue that the path maybe a little bit different from the one currently drawn (you have pointing uncertainties to account for, haven't you?)?

The black dots in Figure 8 (in the revised paper) indicate the DAWN (u,v) *wind profile* vertical locations. The wind profile in turn is created by merging all five LOS beams, each with a different relative viewing direction. These are combined in an optimal way (the ASIA processing referred to in Section 2), and the resulting vertical profile is "placed" at the geographical centroid location from all five beams. Owing to the DAWN conically-located LOS locations, the "location" of the final profile is somewhat arbitrary, but if there are two beams each on either side of the DC-8 flight track, then this location will be somewhere along the aircraft subtrack, but not directly under the DC-8. It represents an aggregation or combination of five different views from different angles, so it really represents some sort of average wind from the air sampled collectively from all five beams. In other words, bin-by-bin comparisons with the APR-2 nadir reflectivity as in Figures 8 needs to take the instrument scan characteristics into account. We have added wording to this effect in the discussion of Figure 8 and the others similar to it.

[Figure]

As for the question on DAWN penetration through clouds. This is something that applies to each of the LOS beams. The radar and lidar systems scan and "stare" very differently. Each DAWN LOS profile is pointed off-nadir 30 degrees at different azimuth angles (Figure 1), where is "stares" over a longer integration time than the APR2 radar (APR2 collects 24 rays as it scans across track in about 1.2 seconds). So, indeed you are absolutely right that DAWN LOS and APR-2 beam matching has bin/beam matching uncertainties associated with it. We did not show this in the manuscript. When each DAWN LOS bin (about 30-m) is mapped to APR2, you end up with a very coarse interpolation. See the figure to the left, which I did not include in the manuscript. Here you can envision the DAWN LOS beam (LOS bin index 1 is the first bin below the DC-8 and bin 320 is the bin at the surface) like a pencil pointing through the APR-2 scanning "volume" covered by its cross-track swath as the DC-8 moves forward. Now imagine the APR-2 cross section along this LOS cross section. Notice how many APR-2 beams are "replicated" (no interpolation was done) owing to the different scan modes of the two instruments. In this case, this LOS beam penetrated an upper cloud portion between 7- and 4-km height, that was sufficiently optically thin enough that DAWN could penetrate it (SNR fell to below 2), but still had dynamic range to capture winds below it (SNR was near 10 near the surface). But, another of the other four LOS beams (of the five total) was unable to penetrate to the near-surface (example not shown). This would be reflected in the quality of the ASIA wind profile processing when it had only four LOS beams to work with. If there were even more cloudy conditions, even less LOS beams are available to retrieve the wind profile. Nonetheless, this picture gives an example of where (in the vertical) the cloud layers are, and how far DAWN could penetrate though the cloud before losing its signal for good.

Sect.4: I understand that the retrieval of wind must be done in the aircraft reference frame for the interpretation it is much better to go back to the usual system (E-W and N-S winds). Since the DC-8 heading is known this is a simple conversion. By so doing you will get rid of all the discussion about the heading and we will actually see the "real winds" (which are the relevant ones for the study of "dynamical processes"). Also the u,v notation is confusing since it is typically used for E-W and N-S winds.

Exactly right. For this example, we intentionally chose the June 11 case since the DC-8 flight bearings were (fortuitously) along E-W and N-S (or vice versa) directions, so this conversion was not necessary. We changed the notation as suggested in the text and Figure 20 (in the revised manuscript). But in general, yes, the aircraft cross-track winds are some mixture of u and v, so the DAWN winds could be transformed (rotated at each level) into the aircraft frame of reference. We think that this June 11 case made the DAWN-APR2 cross-track wind comparison easier to understand.

---

## Author Comment (AC2) · 3 May 2020

**General comments**

This well written manuscript presents novel, unique and relevant collocated airborne Doppler lidar and radar measurements in complex, convective subtropical environments. Focus is set to a common display of both instruments' data sets from two exemplary NASA DC-8 research flights, to show regions of common data overlap and the measurement limits of each instrument. The paper is fully suitable to the scope of AMT, and both the scientific relevance and the data quality are outstanding. However, or because of this, I find it disappointing that scientific conclusions from such an interesting instrument combination are missing, and that the reader is just referred to future studies.

There exists a WCRP Grand Challenge on Clouds, Circulation and Climate Sensitivity (which the authors do not address - anyway), because one of the big science questions is the feedback of convection on dynamics. This DC-8 instrumentation is perfectly suited to address this question, and I think the readers would like to see more details and (preliminary) conclusions hereto.

This manuscript is intentionally weighted almost totally to the observational aspects. As mentioned in the introduction (…*this manuscript will describe and present only the airborne precipitation radar and DWL observations; a separate manuscript will present the associated mesoscale model simulations and DWL data assimilation experiment results*) and in the last paragraph in the conclusions (….*this manuscript provides the observational context for a separate mesoscale model data assimilation study, which is aimed at quantifying the impact of the DAWN measurements on the analyzed atmospheric state variables and on the forecasted precipitation when the DAWN wind profile observations were assimilated into the model*), it is an introductory paper. A Part 2 paper that uses these CPEX data in a data assimilation (DA) study is forthcoming. If the field observations material and the DA study material was submitted as one manuscript, it would be a huge un-focused manuscript. Secondly, there is a technical audience that is mainly interested in the instrumentation and measurement aspects. This paper provides the details for that audience. On the other side of the coin, there is also a large DA community for wind vector DA (evidenced by the increasing number of Aeolus-related papers and presentations), but who already know the basic tenets of Doppler wind lidar and don't necessarily want to know about the CPEX campaign specifics, radar, etc. The second paper will align with that audience. The *Zhang et. al.* (2019) reference provides some indication of what will be in this second paper.

That all said, your comment is a good one and we did add some science related discussion to bridge these two manuscripts. We have mentioned some of our findings from the DA study during the discussion. We added new shear figures that depict the wind shear between the 2- and 8-km and 2- and 6-km levels, which highlight the importance of when and where winds are sampled. In our DA study, we believe that the timing and location of the DAWN observations during the first one-hour period (1830-1930 UTC) captured the wind shear between various mid-upper levels that was not as well defined in the control run, and that its assimilation led to the formation of more convergence and a precipitation pattern that agreed with where APR-2 noted the precipitation. These findings, if substantiated, may be relevant to the WCRP Grand Challenge mentioned, from an observational standpoint. From space, a missing link has been the dynamical aspects. Space based instruments like GPM, and the various passive MW sounders (ATMS, etc.) measure "snapshots" of the condensed water field at some stage in the cloud/convection process, not their motion. We have tried to emphasize the connection between the air motion and the storm development. Capturing motion of air parcels (via a space based lidar) and how they tie into boundary layer processes is indeed a grand challenge.

So here is my suggestion how to avoid disappointment without too much extra effort. The interesting questions are: what is the evolution in time of the probed cloud cluster? Did you observe secondary circulations due to cloud growth? Is convergence or divergence visible in the measurements? So, at the end of section 3, maybe also 4, you should answer these questions. An extra, perhaps 3-d, sketch of the cloud cluster with the essential wind arrows resulting from all flight segments at both 2- and 8-km heights would be very helpful.

Very good suggestion, especially on the last point. Based on your comment and one from a different reviewer, we created 2-level wind hodographs for each of the four one-hour time segments. From these, the directional wind shear (if present, or how it evolves) is more obvious. We did this for the 2-vs-6 km levels and the 2-vs-8 km levels at each of the four one-hour time segments, to capture evolution. Furthermore, I separated this into quadrants relative to the approximate center of this flight box (25.2N 73W). Which show very interesting shear particularly in the 1830-1930 period. During this time there is sustained directional 2-vs-8 km wind shear west of the AOI, oriented from west to east. A similar analysis for the shear between 2-km and 6-km shows the shear oriented more south to north. Data gathered in the two hours after this (2030-2130) (Figure 14 in the revised paper) show that the 2-vs-8 km shear oriented itself more towards the southeast.

[Figure]

On the question of secondary circulations and convergence: From the scale of these observations alone, we can't readily pick out convergence since that requires a larger area and more spatially "complete" field than what these repeat pass, narrow swath DAWN data can provide, especially near the surface. We can say here that the assimilation of the DAWN winds (the Part 2 paper, not yet submitted) which used WRF-like mesoscale model simulations, showed a very discernable impact to wind and precipitation, enhanced surface convergence was noted when DAWN winds were assimilated, as well a more pronounced cold pool, as reflected in the precipitation location and timing.

**Specific comments**

The abstract is misleading. It repeats the nicely written overarching science issues from the introduction (it is OK to address them in the introduction), suggesting to the reader that these big questions are the main topic (which is not quite correct), but in fact it lacks the major results of the paper (= the answers to my above questions). Furthermore, it mentions "transport of water vapor" (line 15) and "Frequent dropsonde data" (l. 19) which both are not major topics of the paper.

Point well taken. We have revised the abstract accordingly.

Lines 37 and 64: these sentences highlight the importance of the vertical distribution of water vapor, a very interesting topic, yet which is (unfortunately) not addressed in this paper. The dropsondes could provide the humidity profiles, but I guess there were too few of them on these two flights to make solid statements, and/or this topic is beyond the scope of this study? Could you comment on this?

Good point. DAWN operates in a spectral band that is away from water vapor absorption. Actually, there is a follow-on CPEX campaign planned (in 2021) that will have, in addition to DAWN and APR-2, a water vapor lidar. From these joint observations, one will be able to depict high resolution water vapor structure together with its dynamical aspects. For this manuscript, we removed the dropsondes discussion since we did not analyze them, and there are many fewer of these compared to DAWN winds (the dropsondes provided important cal-val for the DAWN wind processing).

Line 79: ". . .radar and DWL observations from two exemplary flight days" to be more precise. In this context, I find that Table 1 is not at all needed.

Another purpose of this paper is to document the CPEX field campaign not only for the two dates/cases studied here, but also for future studies by others who may use this paper to pick out other dates that may be more relevant to their particular needs. We feel that Table 1 is important for this purpose.

Fig 1 is difficult to understand. Fig 5 suggests that at 2 km asl the +-45° lidar positions at 30° off-nadir angle have about the same separation than the APR swath width at +-25°. Could that be illustrated in Fig 1?

This figure is designed as a depiction rather than anything angularly or spatially accurate or to scale. A better depiction of the DAWN scanning is in the reference given (*Kavaya et al* 2014). What you say though is approximately correct: The spacing between the ground "footprints" of the DAWN beams at +/-45 degrees is about the same distance (8-km) as the APR-2 radar swath (given a 10-km DC-8 flight altitude). We have made reference to this in the Figure 1 caption.

Line 125: please explain the synoptic situation, and in more detail why you chose this particular situation out of 16 flights. Is the isolated cloud cluster you probed a beginning MCS? What is its relation to the extended cloud band to the north? Was this situation typical for the whole campaign, or was it a very particular "golden day"?

We have added some wording as to why this June 10 date was chosen for analysis. There are three key reasons: (a) This was an isolated, developing set of convective cells, isolated to some degree from large scale forcing effects, (b) the aircraft arrived on-station with sufficient time to make several "box-like" and "crossover" patterns to sample the air mass from different locations, in a developing/growth phase, (c) upon arrival, the clouds during this time were not sufficiently vertically developed and the DC-8 pilot was able to pass over the tops of nearly all clouds during the available on-station time.

Fig 4 is very difficult to understand: what if you would swap Fig 4 and Fig 5? Beginning with the zoom, it would be much easier to understand the heavy full segment 1 overview. Why is the dropsonde at different places? Were there two different dropsondes? Please explain.

We have removed the dropsonde winds in the revised manuscript, and solely focus on the DAWN wind profiles and the APR-2 reflectivity. In answer to your question: The dropsondes were released from the DC-8 at intermittent times, typically after a DC-8 turn maneuver, and where possible away from glaciated clouds where the dropsonde might have iced-up and returned only a partial profile. Whereas DAWN operates on a constant cycle, rotating its optics in the manner mentioned in Figure 1, staring for a period, moving to the next position, etc. DAWN and dropsonde locations will not necessarily coincide. At some times, more dropsondes were released than at others.

Line 225 and caption of Fig 8: explain why you think the convection is growing.

The features noted in the APR-2 Ka-band reflectivity profile near scan 125 indicate a narrow cell with enhanced mid-level reflectivity (ie, maximum is not concentrated in a radar bright band as would be typical of a more stratiform type rainfall phase), and reflectivity starting to push up at or above the approximate 0-degree C freezing level. We (i.e., APR-2 radar operators during CPEX) often noted this feature from the window of the DC-8, a typical picture is shown in Figure 3. On a side note: Note that in Figure 9 (in the revised manuscript) there are 3-4 small cells. Do these organize themselves and develop further, or does the organization fail to hold and the vertical growth dies out? This was one of the questions posed in the original NASA announcement of opportunity (AO) for the CPEX campaign. The DC-8 was not on-station long enough to track this particular evolution. The bigger scientific question of organization of mesoscale convection is largely still an open question, see for example the invited presentation on this by Dr. Ed Zipser (one of the CPEX PI's) from the Fall American Geophysical Union (AGU-2019), available online:

https://agu.confex.com/agu/fm19/meetingapp.cgi/Paper/491084

https://www.youtube.com/watch?v=gbsWSXVwWJc

Fig 15: Is green still the dropsondes? So there are many more dropsondes on this day? Please explain.

We have removed the dropsonde winds in the revised manuscript, and solely focus on the DAWN wind profiles and the APR-2 reflectivity. In answer to your question: green barbs are always meant to refer to dropsonde profiles and red barbs the DAWN wind profiles. DAWN data processing produced 616 profiles for June 10 and 465 profiles for June 11. 26 dropsondes were provided for June 10, and 28 on June 11. On-station times for both dates were not significantly different.

**Technical corrections**

Line 45: 2x "associated", and verb is missing. Fixed.

Line 107: remove "highly capable". Removed.

Line 110: " a constant 30° elevation angle", I do not understand, you mean probably an off-nadir angle of 30°.

DAWN can position its beams in both elevation and azimuth angle. For CPEX, the elevation angle was fixed at 30-degrees, and the azimuth angle was varied. What you say is another way of phrasing the same thing, we have made mention of this.

Fig 2: the GOES image in the expanded box is very coarse, it should be available at much higher resolution if this is a visible imagery, and lat/lon indications would make the big image easier to interpret. Also, highlight the 4 segments from sections 3.1 - 3.4.

The GOES (visible channel) data that is shown in Figure 2 is from the CPEX data portal and unfortunately, we have only these processed data at our disposal. The inset box is somewhat blurred, but it does show the overall zoom-in to the region with sufficient scale to interpret the cloud structure in a qualitative fashion. The DC-8 flight track during each of the four segments are provided on Figures 4, 10, 13 and 16, respectively. If we also show these on Figure 2, it unnecessarily clutters up the figure with detail that is presented anyhow later on. We added wording that the individual flight track segments from Figure 2 are shown right before the beginning of Section 3.1.

Lines 160 and 176: the unit dB is probably wrong when characterizing a DAWN SNR level.

A logarithmic scale (dB) is an appropriate scale for characterizing the SNR in the DAWN LOS signal power.

Line 185: "show the LOS projections to msl from the DC-8" MSL= capitalized. Fixed.

Fig 5: in the upper panel there are no winds, and in the lower panel there is no dropsonde, so you may want to adapt both lower left text boxes. As mentioned, dropsonde analysis was removed. Fixed.

Fig 6: I do not see any dropsonde, so you may want to adapt both lower left text boxes. As mentioned, dropsonde analysis was removed. Fixed.

Line 249: "winds (not shown)", you could refer to Fig 6, showing a region quite close where the winds are shown.

The updated manuscript shows DAWN winds at the 2- and 8-km levels, so this is addressed.

Line 256: "of the DAWN. . ." Fixed.

Figs 9, 11, 13 and 15: the color bar is too large. Fixed.

Line 393: 2x "examine" Fixed.

---

## Author Comment (AC3) · 18 May 2020

Higher resolution GOES-16 visible data have also been now used in the background imagery for ALL figures in the manuscript that had GOES-13 IR data previously. An example of Figure 2 is shown.